# DOMAIN INVARIANT ADVERSARIAL LEARNING

## ABSTRACT

The phenomenon of adversarial examples illustrates one of the most basic vulnerabilities of deep neural networks. Among the variety of techniques introduced to surmount this inherent weakness, adversarial training has emerged as the most effective strategy to achieve robustness. Typically, this is achieved by balancing robust and natural objectives. In this work, we aim to further optimize the trade-off between robust and standard accuracy by enforcing a domain-invariant feature representation. We present a new adversarial training method, *Domain Invariant Adversarial Learning* (DIAL), which learns a feature representation that is both robust and domain invariant. DIAL uses a variant of Domain Adversarial Neural Network (DANN) on the natural domain and its corresponding adversarial domain. In the case where the source domain consists of natural examples and the target domain is the adversarially perturbed examples, our method learns a feature representation constrained not to discriminate between the natural and adversarial examples, and can therefore achieve a more robust representation. Our experiments indicate that our method improves both robustness and standard accuracy, when compared to other state-of-the-art adversarial training methods.

## 1 INTRODUCTION

Deep learning models have achieved impressive success on a wide range of challenging tasks. However, their performance was shown to be brittle to *adversarial examples*: small, imperceptible perturbations in the input that drastically alter the classification (Carlini & Wagner, 2017a;b; Goodfellow et al., 2014; Kurakin et al., 2016b; Moosavi-Dezfooli et al., 2016; Szegedy et al., 2013; Tramèr et al., 2017; Dong et al., 2018; Tabacof & Valle, 2016; Xie et al., 2019b; Rony et al., 2019). Designing reliable robust models has gained significant attention in the arms race against adversarial examples. Adversarial training (Szegedy et al., 2013; Goodfellow et al., 2014; Madry et al., 2017; Zhang et al., 2019b) has been suggested as one of the most effective approaches to defend against such examples, and can be described as solving the following min-max optimization problem:

$$\min_{\theta} \mathbb{E}_{(x,y)\sim\mathcal{D}} \left[ \max_{x':\|x'-x\|_p \leq \epsilon} \ell\left(x', y; \theta\right) \right],$$

where $x'$ is the $\epsilon$-bounded perturbation in the $\ell_p$ norm and $\ell$ is the loss function. Different unrestricted attacks methods were also suggested, such as adversarial deformation, rotations, translation and more (Brown et al., 2018; Engstrom et al., 2018; Xiao et al., 2018; Alaifari et al., 2018; Gilmer et al., 2018).

The resulting min-max optimization problem can be hard to solve in general. Nevertheless, in the context of $\epsilon$-bounded perturbations, the problem is often tractable in practice. The inner maximization is usually approximated by generating adversarial examples using projected gradient descent (PGD) (Kurakin et al., 2016a; Madry et al., 2017). A PGD adversary starts with randomly initialized perturbation and iteratively adjust the perturbation while projecting it back into the $\epsilon$-ball:

$$x_{t+1} = \Pi_{\mathbb{B}_\epsilon(x_0)} \left( x_t + \alpha \cdot \text{sign}(\nabla_{x_t}\ell(G(x_t), y)) \right),$$

where $x_0$ is the natural example (with or without random noise), and $\Pi_{\mathbb{B}_\epsilon(x)}$ is the projection operator onto the $\epsilon$-ball, $G$ is the network, and $\alpha$ is the perturbation step size. As was shown by Athalye et al. (2018), PGD-based adversarial training was one of the few defenses that were not broken under strong attacks.

That said, the gap between robust and natural accuracy remains large for many tasks such as CIFAR-10 (Krizhevsky et al., 2009) and ImageNet (Deng et al., 2009). Generally speaking, Tsipras et al. (2018) suggested that robustness may be at odds with natural accuracy, and usually the trade-off is inherent. Nevertheless, a growing body of work aimed to improve the standard PGD-based adversarial training introduced by Madry et al. (2017) in various ways such as improved adversarial loss functions and regularization techniques (Kannan et al., 2018; Wang et al., 2019b; Zhang et al., 2019b), semi-supervised approaches(Carmon et al., 2019; Uesato et al., 2019; Zhai et al., 2019), adversarial perturbations on model weights (Wu et al., 2020), utilizing out of distribution data (Lee et al., 2021) and many others. See related work for more details.

**Our contribution.** In this work, we propose a novel approach to regulating the tradeoff between robustness and natural accuracy. In contrast to the aforementioned works, our method enhances adversarial training by enforcing a feature representation that is invariant across the natural and adversarial domains. We incorporate the idea of Domain-Adversarial Neural Networks (DANN) (Ganin & Lempitsky, 2015; Ganin et al., 2016) directly into the adversarial training process. DANN is a representation learning approach for domain adaptation, designed to ensure that predictions are made based on invariant feature representation that cannot discriminate between source and target domains. Intuitively, the tasks of adversarial training and of domain-invariant representation have a similar goal: given a source (natural) domain $X$ and a the target (adversarial) domain $X'$, we hope to achieve $g(X) \approx g(X')$, where $g$ a feature representation function (i.e., neural network). Achieving such a dual representation intuitively yields a more general feature representation.

In a comprehensive battery of experiments on MNIST (LeCun et al., 1998), SVHN (Netzer et al., 2011), CIFAR-10 (Krizhevsky et al., 2009) and CIFAR-100 (Krizhevsky et al., 2009) datasets, we demonstrate that by enforcing domain-invariant representation learning using DANN simultaneously with adversarial training, we gain a significant and consistent improvement in both robustness and natural accuracy compared to other state-of-the-art adversarial training methods, under Auto-Attack (Croce & Hein, 2020) and various strong PGD (Madry et al., 2017), CW (Carlini & Wagner, 2017b) adversaries in white-box and black-box settings. Additionally, we evaluate our method using unforeseen "natural" corruptions (Hendrycks & Dietterich, 2018), unforeseen adversaries (e.g., $\ell_1, \ell_2$), transfer learning, and perform ablation studies. Finally, we offer a novel score function for quantifying the robust-natural accuracy tradeoff.

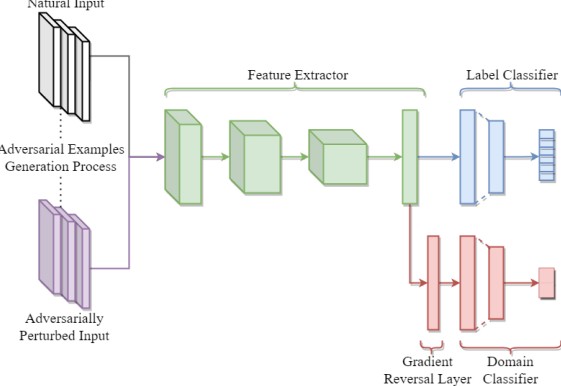

Figure 1: Illustration of the proposed architecture to enforce domain invariant representation. The feature extractor and label classifier form the a regular DNN architecture that can be used for the main natural task. The domain classifier is incorporated alongside the label classifier. The reversal gradient layer multiplies the gradient by a negative number during the back-propagation.

## 2 RELATED WORK

### 2.1 DEFENSE METHODS

A variety of theoretically principled (Cohen et al., 2019; Raghunathan et al., 2018a; Sinha et al., 2017; Raghunathan et al., 2018b; Wong et al., 2018; Wong & Kolter, 2018; Gowal et al., 2018)

and empirical defense approaches (Bai et al., 2021) were proposed to enhance robustness since the discovery of adversarial examples. Among the empirical defence techniques we can find adversarial regularization (Kurakin et al., 2016a; Madry et al., 2017; Zhang et al., 2019b; Wang et al., 2019b; Kannan et al., 2018), curriculum-based adversarial training (Cai et al., 2018; Zhang et al., 2020; Wang et al., 2019a), ensemble adversarial training (Tramèr et al., 2017; Pang et al., 2019; Yang et al., 2020), adversarial training with adaptive attack budget (Ding et al., 2018; Cheng et al., 2020), semi-supervised and unsupervised adversarial training (Carmon et al., 2019; Uesato et al., 2019; Zhai et al., 2019), robust self/pre-training (Jiang et al., 2020; Chen et al., 2020), efficient adversarial training (Shafahi et al., 2019; Wong et al., 2020; Andriushchenko & Flammarion, 2020; Zhang et al., 2019a), and many other techniques (Zhang & Wang, 2019; Goldblum et al., 2020; Pang et al., 2020b; Lee et al., 2020). In an additional research direction, researchers suggested to add new dedicated building blocks to the network architecture for improved robustness (Xie & Yuille, 2019; Xie et al., 2019a; Liu et al., 2020). Liu et al. (2020) hypothesised that different adversaries belong to different domains, and suggested gated batch normalization which is trained with multiple perturbation types. Others focused on searching robust architectures against adversarial examples (Guo et al., 2020).

Our work belongs to the the family of adversarial regularization techniques, for which we elaborate on common and best performing methods, and highlight the differences compared to our method.

Madry et al. (2017) proposed a technique, commonly referred to as Adversarial Training (AT), to minimize the cross entropy loss on adversarial examples generated by PGD. Zhang et al. (2019b) suggested to decompose the prediction error for adversarial examples as the sum of the natural error and boundary error, and provided a differentiable upper bounds on both terms. Motivated by this decomposition, they suggested a technique called TRADES that uses the Kullback-Leibler (KL) divergence as a regularization term that will push the decision boundary away from the data. Wang et al. (2019b) suggested that misclassified examples have a significant impact on final robustness, and proposed a technique called MART that differentiate between correctly classified and miss-classified examples during training.

Another area of research aims at revealing the connection between the loss weight landscape and adversarial training (Prabhu et al., 2019; Yu et al., 2018; Wu et al., 2020). Specifically, Wu et al. (2020) identified a correlation between the flatness of weight loss landscape and robust generalization gap. They proposed the Adversarial Weight Perturbation (AWP) mechanism that is integrated into existing adversarial training methods. More recently, this approach was formalized from a theoretical standpoint by Tsai et al. (2021). However, this method forms a double-perturbation mechanism that perturbs both inputs and weights, which may incur a significant increase in calculation overhead. Nevertheless, we show that DIAL still improves state-of-the-art results when combined with AWP.

A related approach to ours, called ATDA, was presented by Song et al. (2018). They proposed to add several constrains to the loss function in order to enforce domain adaptation: correlation alignment and maximum mean discrepancy (Borgwardt et al., 2006; Sun & Saenko, 2016). While the objective is similar, using ideas from domain adaptation for learning better representation, we address it in two different ways. Our method fundamentally differs from Song et al. (2018) since we do not enforce domain adaptation by adding specific constrains to the loss function. Instead, we let the network learn the domain invariant representation directly during the optimization process, as suggested by Ganin & Lempitsky (2015); Ganin et al. (2016). Moreover, Song et al. (2018) focused mainly of FGSM. We empirically demonstrate the superiority of our method in Section 4. In a concurrent work, Qian et al. (2021) utilized the idea of exploiting local and global data information, and suggested to generate the adversarial examples by attacking an additional domain classifier.

## 2.2 ROBUST GENERALIZATION

Several works investigated the sample complexity requires the ensure adversarial generalization compared to the non-adversarial counterpart. Schmidt et al. (2018) has shown that there exists a distribution (mixture of Gaussians) where ensuring robust generalization necessarily requires more data than standard learning. This has been furthered investigated in a distribution-free models via the Rademacher Complexity, VC-dimension (Yin et al., 2019; Attias et al., 2019; Khim & Loh, 2018; Awasthi et al., 2020; Cullina et al., 2018; Montasser et al., 2019; Tsai et al., 2021) and additional settings (Diochnos et al., 2018; Carmon et al., 2019).

## 3 Domain Invariant Adversarial Learning approach

In this section, we introduce our Domain Invariant Adversarial Learning (DIAL) approach for adversarial training. The source domain is the natural dataset, and the target domain is generated using an adversarial attack on the natural domain. We aim to learn a model that has low error on the source (natural) task (e.g., classification) while ensuring that the internal representation cannot discriminate between the natural and adversarial domains. In this way, we enforce additional regularization on the feature representation, which enhances the robustness.

### 3.1 Model architecture and regularized loss function

Let us define the notation for our domain invariant robust architecture and loss. Let $G_f(\cdot; \theta_f)$ be the feature extractor neural network with parameters $\theta_f$. Let $G_y(\cdot; \theta_y)$ be the label classifier with parameters $\theta_y$, and let $G_d(\cdot; \theta_d)$ the domain classifier with parameters $\theta_d$. That is, $G_y(G_f(\cdot; \theta_f); \theta_y)$ is essentially the standard model (e.g., wide residual network (Zagoruyko & Komodakis, 2016)), while in addition, we have a domain classification layer to enforce a domain invariant on the feature representation. An illustration of the architecture is presented in Figure 1.

Given a training set $\{(x_i, y_i)\}_{i=1}^n$, the natural loss is defined as:

$$\mathcal{L}_{\text{nat}}^y = \tfrac{1}{n} \sum_{i=1}^n \text{CE}(G_y(G_f(x_i; \theta_f); \theta_y), y_i).$$

We consider two basic forms of the robust loss. One is the standard cross-entropy (CE) loss between the predicted probabilities and the actual label, which we refer to later as $\text{DIAL}_{\text{CE}}$. The second is the Kullback-Leibler (KL) divergence between the adversarial and natural model outputs (logits), as in Zhang et al. (2019b); Wang et al. (2019b), which we refer to as $\text{DIAL}_{\text{KL}}$.

$$\mathcal{L}_{\text{rob}}^{\text{CE}} = \tfrac{1}{n} \sum_{i=1}^n \text{CE}(G_y(G_f(x_i'; \theta_f); \theta_y), y_i),$$
$$\mathcal{L}_{\text{rob}}^{\text{KL}} = \tfrac{1}{n} \sum_{i=1}^n \text{KL}(G_f(x_i'; \theta_f) \parallel G_f(x_i; \theta_f)),$$

where $\{(x_i', y_i)\}_{i=1}^n$ are the generated corresponding adversarial examples. Next, we define source domain label $d_i$ as 0 (for natural examples) and target domain label $d_i'$ as 1 (for adversarial examples). Then, the natural and adversarial domain losses are defined as:

$$\mathcal{L}_{\text{nat}}^d = \tfrac{1}{n} \sum_{i=1}^n \text{CE}(G_d(G_f(x_i; \theta_f); \theta_d), d_i),$$
$$\mathcal{L}_{\text{adv}}^d = \tfrac{1}{n} \sum_{i=1}^n \text{CE}(G_d(G_f(x_i'; \theta_f); \theta_d), d_i').$$

We can now define the full domain invariant robust loss:

$$\text{DIAL}_{\text{CE}} = \mathcal{L}_{\text{nat}}^y + \lambda \mathcal{L}_{\text{rob}}^{\text{CE}} - r(\mathcal{L}_{\text{nat}}^d + \mathcal{L}_{\text{adv}}^d),$$
$$\text{DIAL}_{\text{KL}} = \mathcal{L}_{\text{nat}}^y + \lambda \mathcal{L}_{\text{rob}}^{\text{KL}} - r(\mathcal{L}_{\text{nat}}^d + \mathcal{L}_{\text{adv}}^d).$$

The goal is to *minimize* the loss on the natural and adversarial classification while *maximizing* the loss for the domains. The *reversal-ratio* hyper-parameter $r$ is inserted into the network layers as a gradient reversal layer (Ganin & Lempitsky, 2015; Ganin et al., 2016) that leaves the input unchanged during forward propagation and reverses the gradient by multiplying it with a negative scalar during the back-propagation. The reversal-ratio parameter is initialized to a small value and is gradually increased to $r$, as the main objective converges. This enforces a domain-invariant representation as the training progress: a larger value enforces a higher fidelity to the domain. A comprehensive algorithm description can be found in Appendix A.

**Modularity and semi-supervised extensions.** We note that the domain classifier is a modular component that can be integrated into existing models for further improvements. Moreover, since the domain classifier does not require the class labels, additional unlabeled data can be leveraged in future work for improved results.

### 3.2 The benefits of invariant representation to adversarial examples

The motivation behind the proposed method is to enforce an invariant feature representation to adversarial perturbations. Given a natural example $x$ and its adversarial counterpart $x'$, if the domain classifier manages to distinguish between them, this means that the perturbation has induced a significant difference in the feature representation. We impose an additional loss on the natural and adversarial domains in order to discourage this behavior.

We demonstrate that the feature representation layer does not discriminate between natural and adversarial examples, namely $G_f(x; \theta_f) \approx G_f(x'; \theta_f)$. Figure 2 presents the scaled mean and standard deviation (std) of the absolute differences between the natural examples from test and their corresponding adversarial examples on different features from the feature representation layer. Smaller differences in the mean and std imply a higher domain invariance — and indeed, DIAL achieves near-zero differences almost across the board. Moreover, DIAL's feature-level invariance almost consistently outperforms the naturally trained model and the model trained using standard adversarial training techniques (Madry et al., 2017). We provide additional features visualizations in Appendix H.

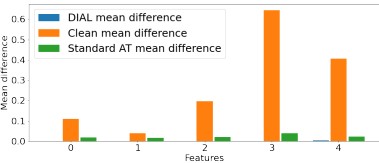
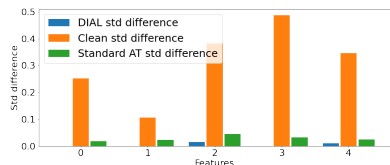

| (a) Mean difference comparison | (b) Standard deviation difference comparison |

Figure 2: Mean and std differences comparison between DIAL, naturally trained model and model trained using standard adversarial training on five random features from the features layer. Each bar represent the difference between the means/std of the natural examples and the mean/std of their corresponding adversarial examples on this same feature.

## 4 Experiments

In this section we conduct comprehensive experiments to emphasise the effectiveness of DIAL, including evaluations under white-box and black-box settings, robustness to unforeseen adversaries, robustness to unforeseen corruptions, transfer learning, and ablation studies. Finally, we present a new measurement to test the balance between robustness and natural accuracy, which we named $F_1$-robust score.

### 4.1 A case study on SVHN and CIFAR-100

In the first part of our analysis, we conduct a case study experiment on two benchmark datasets: SVHN (Netzer et al., 2011) and CIFAR-100 Krizhevsky et al. (2009). We follow common experiment settings as in Rice et al. (2020); Wu et al. (2020). We used the PreAct ResNet-18 (He et al., 2016) architecture on which we integrate a domain classification layer. The adversarial training is done using 10-step PGD adversary with perturbation size of $\epsilon = 0.031$ and a step size of 0.003 for SVHN and 0.007 for CIFAR-100. The batch size is 128, weight decay is $7e^{-4}$ and the model is trained for 100 epochs. For SVHN, the initial learinnig rate is set to 0.01 and decays by a factor of 10 after 55, 75 and 90 iteration. For CIFAR-100, the initial learning rate is set to 0.1 and decays by a factor of 10 after 75 and 90 iterations. Results are averaged over 3 restarts while omitting one standard deviation. As can be seen by the results in Tables 1 and 2, DIAL presents consistent improvement in robustness (e.g., 5.75% improved robustness on SVHN against AA) compared to the standard AT while also improving the natural accuracy. More results are presented in Appendix B.

### 4.2 Benchmarking the State-of-the-art Robustness

In this part, we evaluate the performance of DIAL compared to other state-of-the-art methods on CIFAR-10. We follow the same experiment setups as in Madry et al. (2017); Wang et al. (2019b);

Table 1: Robustness against white-box, black-box attacks and Auto-Attack (AA) on SVHN. Black-box attacks are generated using naturally trained surrogate model.

| Defense Model | Natural | White-box | | | | Black-Box | | | | AA |
| | | $PGD^{20}$ | $PGD^{100}$ | $PGD^{1000}$ | $CW^{\infty}$ | $PGD^{20}$ | $PGD^{100}$ | $PGD^{1000}$ | $CW^{\infty}$ | |
| --- | --- | --- | --- | --- | --- | --- | --- | --- | --- | --- |
| AT | 89.90 | 53.23 | 49.45 | 49.23 | 48.25 | 86.44 | 86.28 | 86.18 | 86.42 | 45.25 |
| $DIAL_{KL}$ (Ours) | 90.66 | **58.91** | **55.30** | **55.11** | **53.67** | 87.62 | 87.52 | 87.41 | 87.63 | **51.00** |
| $DIAL_{CE}$ (Ours) | **92.88** | 55.26 | 50.82 | 50.54 | 49.66 | **89.12** | **89.01** | **88.74** | **89.10** | 46.52 |

Table 2: Robustness against white-box, black-box attacks and Auto-Attack (AA) on CIFAR100. Black-box attacks are generated using naturally trained surrogate model.

| Defense Model | Natural | White-box | | | | Black-Box | | | | AA |
| | | $PGD^{20}$ | $PGD^{100}$ | $PGD^{1000}$ | $CW^{\infty}$ | $PGD^{20}$ | $PGD^{100}$ | $PGD^{1000}$ | $CW^{\infty}$ | |
| --- | --- | --- | --- | --- | --- | --- | --- | --- | --- | --- |
| AT | 56.73 | 29.57 | 28.45 | 28.39 | 26.6 | 55.52 | 55.29 | 55.26 | 55.40 | 24.12 |
| $DIAL_{KL}$ (Ours) | 58.47 | **31.19** | **30.50** | **30.42** | **26.91** | 57.16 | 56.81 | 56.80 | 57.00 | **25.87** |
| $DIAL_{CE}$ (Ours) | **60.77** | 27.87 | 26.66 | 26.61 | 25.98 | **59.48** | **59.06** | **58.96** | **59.20** | 23.51 |

Zhang et al. (2019b). When experiment settings are not identical between tested methods, we choose the most commonly used settings, and apply it to all experiments. This way, we keep the comparison as fair as possible and avoid reporting changes in results which are caused by inconsistent experiment settings (Pang et al., 2020a). To show that our results are not caused because of what is referred to as *obfuscated gradients* (Athalye et al., 2018), we evaluate our method with same setup as in our defense model, under strong attacks (e.g., $PGD^{1000}$) in both white-box, black-box settings, Auto-Attack (Croce & Hein, 2020), unforeseen "natural" corruptions (Hendrycks & Dietterich, 2018), and unforeseen adversaries. To make sure that the reported improvements are not caused by *adversarial overfitting* (Rice et al., 2020), we report best robust results for each method on average of 3 restarts, while omitting one standard deviation. Additional results for CIFAR-10 as well as comprehansive evaluation on MNIST can be found in Appendix D and E.

**CIFAR-10 setup.** We use the wide residual network (WRN-34-10) (Zagoruyko & Komodakis, 2016) architecture. Sidelong this architecture, we integrate a domain classification layer. To generate the adversarial domain dataset, we use a perturbation size of $\epsilon = 0.031$. We apply 10 of inner maximization iterations with perturbation step size of 0.007. Batch size is set to 128, weight decay is set to $7e^{-4}$, and the model is trained for 100 epochs. Similar to the other methods, the initial learning rate was set to 0.1, and decays by a factor of 10 at iteration 75 and 90. We also introduce a version of our method that incorporates the AWP double-perturbation mechanism, named DIAL-AWP. For black-box attacks, we used two types of surrogate models (1) surrogate model trained independently without adversarial training, with natural accuracy of 95.61% and (2) surrogate model trained using one of the adversarial training methods. Additional training details can be found in Appendix C.

**White-box/Black-box robustness.** As reported in Table 3, our method achieves better robustness over the other state-of-the-art methods with respect to the different attacks. Specifically, in white-box settings, we see that our method improves robustness over Madry et al. (2017) by more than 2%, and roughly 2% over TRADES using the common $PGD^{20}$ attack while keeping higher natural accuracy. We also observe better natural accuracy of 1.65% over MART while also achieving better robustness over all attacks. Moreover, our method presents significant improvement of up to 15% compared to the the domain invariant method suggested by Song et al. (2018) (ATDA). When incorporating AWP, our method improves the TRADES-AWP variant by almost 2% Additional results are available in Appendix E. When tested on black-box settings, $DIAL_{CE}$ presents a significant improvement of more than 4.4% over the second-best performing method, and up to 13%. In Table 4, we also present the black-box results when the source model is taken from one of the adversarially trained models. In addition to the improvement in black-box robustness, $DIAL_{CE}$ also manages to achieve better clean accuracy of more than 4.5% over the second-best performing method. Moreover, based on the

auto-attack leader-board [1], our method achieves the 1st place among models without additional data using the WRN-34-10 architecture.

Table 3: Robustness against white-box, black-box attacks and Auto-Attack (AA) on CIFAR-10. Black-box attacks are generated using naturally trained surrogate model.

| Defense Model | Natural | White-box | | | Black-Box | | | |
| | | $PGD^{20}$ | $PGD^{100}$ | $CW^{\infty}$ | $PGD^{20}$ | $PGD^{100}$ | $CW^{\infty}$ | AA |
|---|---|---|---|---|---|---|---|---|
| TRADES | 84.92 | 56.60 | 55.56 | 54.20 | 84.08 | 83.89 | 83.91 | 53.08 |
| MART | 83.62 | 58.12 | 56.48 | 53.09 | 82.82 | 82.52 | 82.80 | 51.10 |
| AT | 85.10 | 56.28 | 54.46 | 53.99 | 84.22 | 84.14 | 83.92 | 51.52 |
| ATDA | 76.91 | 43.27 | 41.13 | 41.01 | 75.59 | 75.37 | 75.35 | 40.08 |
| $DIAL_{KL}$ (Ours) | 85.25 | **58.43** | **56.80** | **55.00** | 84.30 | 84.18 | 84.05 | **53.75** |
| $DIAL_{CE}$ (Ours) | **89.59** | 54.31 | 51.67 | 52.04 | **88.60** | **88.39** | **88.44** | 49.85 |
| $DIAL_{AWP}$ (Ours) | **85.91** | **61.10** | **59.86** | **57.67** | **85.13** | **84.93** | **85.03** | **56.78** |
| $TRADES_{AWP}$ | 85.36 | 59.27 | 59.12 | 57.07 | 84.58 | 84.58 | 84.59 | 56.17 |

Table 4: Black-box $PGD^{20}$ attack using the adversarially trained surrogate models on CIFAR-10.

| Surrogate (source) model | Target model | robustness % |
|---|---|---|
| TRADES | $DIAL_{CE}$ | **67.77** |
| $DIAL_{CE}$ | TRADES | 65.75 |
| MART | $DIAL_{CE}$ | **70.30** |
| $DIAL_{CE}$ | MART | 64.91 |
| AT | $DIAL_{CE}$ | **65.32** |
| $DIAL_{CE}$ | AT | 63.54 |
| ATDA | $DIAL_{CE}$ | **66.77** |
| $DIAL_{CE}$ | ATDA | 52.56 |

### 4.2.1 ROBUSTNESS TO UNFORESEEN ATTACKS AND CORRUPTIONS

**Unforeseen Adversaries.** To further demonstrate the effectiveness of our approach, we test our method against various adversaries that were not used during the training process. We attack the model under the white-box settings with $\ell_2$-PGD, $\ell_1$-PGD, $\ell_\infty$-DeepFool and $\ell_2$-DeepFool (Moosavi-Dezfooli et al., 2016) adversaries using Foolbox (Rauber et al., 2017). We applied commonly used attack budget (perturbation for PGD adversaries and overshot for DeepFool adversaries) with 20 iterations for the PGD adversaries and 50 for the DeepFool adversaries. Results are presented in Table 5. As can be seen by the results, our approach gains an improvement of up to 4.73% over the second best method under the various attack types and average improvement of 3.7% over all threat models.

Table 5: Robustness on CIFAR-10 against unseen adversaries under white-box settings.

| Threat Model | Attack Constraints | $DIAL_{KL}$ | $DIAL_{CE}$ | AT | TRADES | MART | ATDA |
|---|---|---|---|---|---|---|---|
| $\ell_2$-PGD | $\epsilon = 0.5$ | 76.05 | **80.51** | 76.82 | 76.57 | 75.07 | 66.25 |
| | $\epsilon = 0.25$ | 80.98 | **85.38** | 81.41 | 81.10 | 80.04 | 71.87 |
| $\ell_1$-PGD | $\epsilon = 12$ | 74.84 | **80.00** | 76.17 | 75.52 | 75.95 | 65.76 |
| | $\epsilon = 7.84$ | 78.69 | **83.62** | 79.86 | 79.16 | 78.55 | 69.97 |
| $\ell_2$-DeepFool | overshoot=0.02 | 84.53 | **88.88** | 84.15 | 84.23 | 82.96 | 76.08 |
| $\ell_\infty$-DeepFool | overshoot=0.02 | 68.43 | **69.50** | 67.29 | 67.60 | 66.40 | 57.35 |

---

[1]https://github.com/fra31/auto-attack

**Unforeseen Corruptions.** We further demonstrate that our method consistently holds against unforeseen "natural" corruptions, consists of 18 unforeseen diverse corruption types proposed by Hendrycks & Dietterich (2018) on CIFAR-10, which we refer to as CIFAR10-C. The CIFAR10-C benchmark covers noise, blur, weather, and digital categories. As can be shown in Figure 3, our method gains a significant and consistent improvement over all the other methods. Our approach leads to an average improvement of 4.7% with minimum improvement of 3.5% and maximum improvement of 5.9% compared to the second best method over all unforeseen attacks. See Appendix F for the full experiment results.

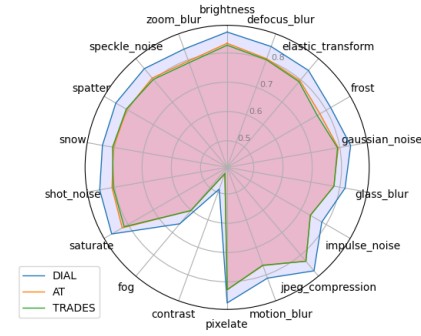

Figure 3: Accuracy comparison over all unforeseen corruptions.

### 4.2.2 TRANSFER LEARNING

Recent works (Salman et al., 2020; Utrera et al., 2020) suggested that robust models transfer better on standard downstream classification tasks. In Table 6 we demonstrate the advantage of our method when applied for transfer learning across CIFAR10 and CIFAR100 using the common linear evaluation protocol. see Appendix G for detailed experiment settings.

Table 6: Transfer learning results comparison.

| Source | Defence Model | Target | |
|---|---|---|---|
| | | CIFAR10 | CIFAR100 |
| CIFAR10 | DIAL | | **28.57** |
| | AT | - | 26.95 |
| | TRADES | | 25.40 |
| CIFAR100 | DIAL | **73.68** | |
| | AT | 71.41 | - |
| | TRADES | 71.42 | |

### 4.2.3 ABLATION STUDIES

In this part, we conduct ablation studies to further investigate the contribution of the additional domain head component introduced in our method. Experiment configuration are as in 4.2, and robust accuracy is reported on white-box PGD[20]. We use the CIFAR-10 dataset and train WRN-34-10. We remove the domain head from both $DIAL_{KL}$ and $DIAL_{CE}$ (equivalent to $r = 0$) and report the natural and robust accuracy. We perform 3 random restarts and omit one standard deviation from the results. Results are presented in Figure 4. Both DIAL variants exhibits stable improvements on both natural accuracy and robust accuracy. $DIAL_{CE}$ and $DIAL_{KL}$ present an improvement of 1.82% and 0.33% on natural accuracy and 2.5% and 1.87% on robust accuracy, respectively.

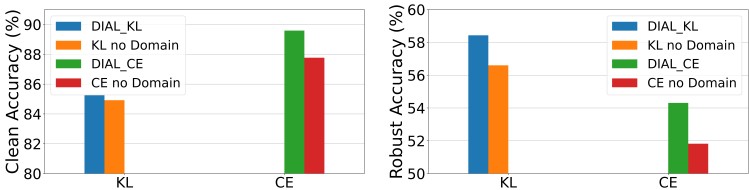

Figure 4: Ablation studies for $DIAL_{KL}$ and $DIAL_{CE}$ on CIFAR-10.

### 4.2.4 VISUALIZING DIAL

To further illustrate our method, we visualize the model outputs using the different methods under natural test data and adversarial test data generated using PGD[20] white-box attack with step size 0.003 and $\epsilon = 0.031$ on CIFAR-10. Figure 5 shows the embedding received after applying t-SNE (Van der Maaten & Hinton, 2008) with two components on the model output for our method and for TRADES. DIAL seems to preserve strong separation between classes on both natural test data and adversarial test data. Additional illustrations for the other methods are attached in Appendix H.

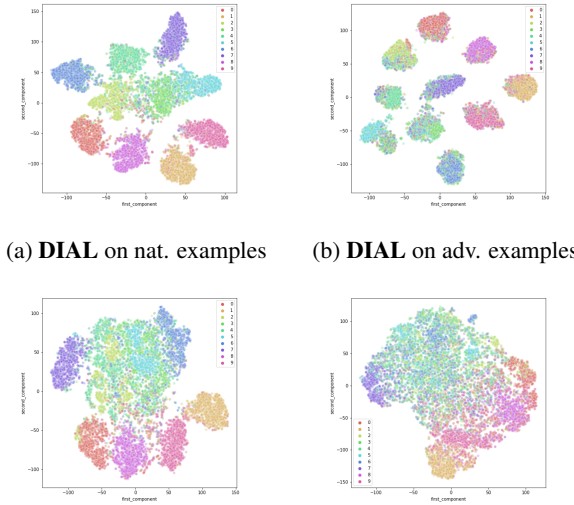

(a) **DIAL** on nat. examples  (b) **DIAL** on adv. examples

(c) **TRADES** on nat. examples (d) **TRADES** on adv. examples

Figure 5: t-SNE embedding of model output (logits) into two-dimensional space for DIAL and TRADES using the CIFAR-10 natural test data and the corresponding adversarial examples.

### 4.3 BALANCED MEASUREMENT FOR ROBUST AND NATURAL ACCURACY

One of the goals of our method is to better balance between robust and natural accuracy under a given model. For a balanced metric, we adopt the idea of $F_1$-score, which is the harmonic mean between the precision and recall. However, rather than using precision and recall, we measure the $F_1$-score between robustness and natural accuracy, using a measure we call the **$F_1$-robust** score.

$$F_1\text{-robust} = \frac{\text{true\_robust}}{\text{true\_robust} + \frac{1}{2}(\text{false\_robust} + \text{false\_natural})},$$

where true_robust are the adversarial examples that were correctly classified, false_robust are the adversarial examples that where miss-classified, and false_natural are the natural examples that were miss-classified. We tested the proposed $F_1$-robust score using PGD[20] on CIFAR-10 dataset in white-box and black-box settings. Results are presented in Table 7 and show that our method achieves the best $F_1$-robust score in both settings, which supports our findings from previous sections.

Table 7: $F_1$-robust measurement using PGD[20] attack in white and black box settings on CIFAR-10.

|  | TRADES | MART | AT | ATDA | DIAL$_{CE}$ | DIAL$_{KL}$ | DIAL$_{AWP}$ | TRADES$_{AWP}$ |
|---|---|---|---|---|---|---|---|---|
| White-box | 0.659 | 0.666 | 0.657 | 0.518 | 0.660 | **0.675** | **0.698** | 0.682 |
| Black-box | 0.844 | 0.831 | 0.845 | 0.761 | **0.890** | 0.847 | **0.854** | 0.849 |

## 5 CONCLUSION

In this paper, we investigated the hypothesis that domain invariant representation can be beneficial for robust learning. With this idea in mind, we proposed a new adversarial learning method, called *Domain Invariant Adversarial Learning* (DIAL) that incorporates Domain Adversarial Neural Network into the adversarial training process. The proposed method is generic and can be combined with any network architecture in a wide range of tasks. Our evaluation process included strong adversaries , unforeseen adversaries , unforeseen corruptions, transfer learning tasks, and ablation studies. Using the extensive empirical analysis, we demonstrate the significant and consistent improvement obtained by DIAL in both robustness and natural accuracy compared to other defence methods on benchmark datasets.

ETHICS STATEMENT

We proposed DIAL to improve models' robustness against adversarial attacks. We hope that it will help in building more secure models for real-world applications. DIAL is comparable to the state-of-the-art methods we tested in terms of training times and other resources. That said, this work is not without limitations: adversarial training is still a computationally expensive procedure that requires extra computations compared to standard training, with the concomitant environmental costs. Even though our method introduced better standard accuracy, adversarial training still degrades the standard accuracy. Moreover, models are trained to be robust using well known threat models such as the bounded $\ell_p$ norms. However, once a model is deployed, we cannot control the type of attacks it faces from sophisticated adversaries. Thus, the general problem is still very far from being fully solved.

REPRODUCIBILITY STATEMENT

In this paper, great efforts were made to ensure that comparison is fair, and all necessary information for reproducibility is present. Section 4 and Appendix B and C contains all experiment settings for SVHN, CIFAR-10 and CIFAR-100 experiments. Appendix D contains all experiment details and results for MNIST experiments. Appendix G contains experiment settings for the transfer learning experiment. In the supplementary material, we provided the source code to train and evaluate DIAL.

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

## A  DOMAIN INVARIANT ADVERSARIAL LEARNING ALGORITHM

Algorithm 1 describes a pseudo-code of our proposed $\text{DIAL}_{\text{CE}}$ variant. As can be seen, a target domain batch is not given in advance as with standard domain-adaptation task. Instead, for each natural batch we generate a target batch using adversarial training. The loss function is composed of natural and adversarial losses with respect to the main task (e.g., classification), and from natural and adversarial domain losses. By maximizing the losses on the domain we aim to learn a feature representation which is invariant to the natural and adversarial domain, and therefore more robust.

---

**Algorithm 1:** Domain Invariant Adversarial Learning

**Input:** Source data $S = \{(x_i, y_i)\}_{i=1}^n$ and network architecture $G_f, G_y, G_d$

**Parameters:** Batch size $m$, perturbation size $\epsilon$, pgd attack step size $\tau$, adversarial trade-off $\lambda$,
      initial reversal ratio $r$, and step size $\alpha$

**Initialize:** $Y_0$ and $Y_1$ source and target domain vectors filled with 0 and 1 respectively

**Output:** Robust network $G = (G_f, G_y, G_d)$ parameterized by $\hat{\theta} = (\theta_f, \theta_y, \theta_d)$ respectively

**1 while** *user-specified stopping criterion is not met* **do**

**2**   Fetch mini-batch $X_s = \{x_j\}_{j=1}^m$, $Y_s = \{y_j\}_{j=1}^m$

**3**   # Generate adversarial target domain batch $X_t$

**4**   **for** $j = 1, \ldots, m$ *(in parallel)* **do**

**5**    $x'_j \leftarrow PGD(x_j, y_j, \epsilon, \tau)$

**6**    $X_t \leftarrow X_t + x'_j$

**7**   **end**

**8**   $\ell_s^y, \ell_t^y \leftarrow \text{CE}(G_y(G_f(X_s)), Y_s), \text{CE}(G_y(G_f(X_t)), Y_s)$

**9**   $\ell_s^d, \ell_t^d \leftarrow \text{CE}(G_d(G_f(X_s)), Y_0), \text{CE}(G_d(G_f(X_t)), Y_1)$

**10**   $\ell \leftarrow \ell_s^y + \lambda\ell_t^y - r(\ell_s^d + \ell_t^d)$

**11**   $\hat{\theta} \leftarrow \hat{\theta} - \alpha\nabla_{\hat{\theta}}(\ell)$

**12 end**

---

## B  ADDITIONAL RESULTS ON CIFAR-100 AND SVHN

Table 8: Robustness against white-box, black-box attacks and Auto-Attack (AA) on SVHN. Black-box attacks are generated using naturally trained surrogate model and applied to the best performing robust models.

| Defense Model | Natural | White-box | | | | Black-Box | | | | AA |
|---|---|---|---|---|---|---|---|---|---|---|
| | | $\text{PGD}^{20}$ | $\text{PGD}^{100}$ | $\text{PGD}^{1000}$ | $\text{CW}^{\infty}$ | $\text{PGD}^{20}$ | $\text{PGD}^{100}$ | $\text{PGD}^{1000}$ | $\text{CW}^{\infty}$ | |
| TRADES | 90.35 | 57.10 | 54.13 | 54.08 | 52.19 | 86.89 | 86.73 | 86.57 | 86.70 | 49.5 |
| $\text{DIAL}_{\text{KL}}$ (Ours) | 90.66 | **58.91** | **55.30** | **55.11** | **53.67** | 87.62 | 87.52 | 87.41 | 87.63 | **51.00** |
| $\text{DIAL}_{\text{CE}}$ (Ours) | **92.88** | 55.26 | 50.82 | 50.54 | 49.66 | **89.12** | **89.01** | **88.74** | **89.10** | 46.52 |

Table 9: Robustness against white-box, black-box attacks and Auto-Attack (AA) on CIFAR100. Black-box attacks are generated using naturally trained surrogate model and applied to the best performing robust models.

| Defense Model | Natural | White-box | | | | Black-Box | | | | AA |
|---|---|---|---|---|---|---|---|---|---|---|
| | | $PGD^{20}$ | $PGD^{100}$ | $PGD^{1000}$ | $CW^{\infty}$ | $PGD^{20}$ | $PGD^{100}$ | $PGD^{1000}$ | $CW^{\infty}$ | |
| TRADES | 58.24 | 30.1 | 29.66 | 29.64 | 25.97 | 57.05 | 56.71 | 56.67 | 56.77 | 24.92 |
| $DIAL_{KL}$ (Ours) | 58.47 | **31.19** | **30.50** | **30.42** | **26.91** | 57.16 | 56.81 | 56.80 | 57.00 | **25.87** |
| $DIAL_{CE}$ (Ours) | **60.77** | 27.87 | 26.66 | 26.61 | 25.98 | **59.48** | **59.06** | **58.96** | **59.20** | 23.51 |

## C  CIFAR-10 ADDITIONAL EXPERIMENTAL SETUP DETAILS

**Additional defence setup.**   For being consistent with other methods, the natural images are padded with 4-pixel padding with 32-random crop and random horizontal flip. Furthermore, all methods are trained using SGD with momentum 0.9. For $DIAL_{KL}$, we balance the robust loss with $\lambda = 6$ and the domains loss with $r = 4$. For $DIAL_{CE}$, we balance the robust loss with $\lambda = 1$ and the domains loss with $r = 2$. For DIAL-AWP, we used the same learning rate schedule used in Wu et al. (2020), where the initial 0.1 learning rate decays by a factor of 10 after 100 and 150 iterations.

## D  BENCHMARKING THE STATE-OF-THE-ART ON MNIST

**Defence setup.**   We use the same CNN architecture as used in Zhang et al. (2019b) which consists of four convolutional layers and three fully-connected layers. Sidelong this architecture, we integrate a domain classification layer. To generate the adversarial domain dataset, we use a perturbation size of $\epsilon = 0.3$. We apply 40 iterations of inner maximization with perturbation step size of 0.01. Batch size is set to 128 and the model is trained for 100 epochs. Similar to the other methods, the initial learning rate was set to 0.01, and decays by a factor of 10 after 55 iterations, 75 and 90 iterations. All the models in the experiment are trained using SGD with momentum 0.9. For our method, we balance the robust loss with $\lambda = 6$ and the domains loss with $r = 0.1$.

**White-box/Black-box robustness.**   We evaluate all defense models using $PGD^{40}$, $PGD^{100}$, $PGD^{1000}$ and $CW_{\infty}$ ($\ell_{\infty}$ version of Carlini & Wagner (2017b) attack optimized by PGD-100) with step size 0.01. We constrain all attacks by the same perturbation $\epsilon = 0.3$. For our black-box setting, we use a naturally trained surrogate model with natural accuracy of 99.51%. As reported in Table 10, our method achieves improved robustness over the other methods under the different attack types, while preserving the same level of natural accuracy, and even surpassing the naturally trained model. We should note that in general, the improvement margin on MNIST is more moderate compared to CIFAR-10, since MNIST is an easier task than CIFAR-10 and the robustness range is already high to begin with. Additional results are available in Appendix E.

Table 10: Robustness against white-box, black-box attacks and Auto-Attack (AA) on MNIST. Black-box attacks are generated using naturally trained surrogate model and applied to the best performing robust models.

| Defense Model | Natural | White-box | | | Black-Box | | | AA |
|---|---|---|---|---|---|---|---|---|
| | | $PGD^{40}$ | $PGD^{100}$ | $CW^{\infty}$ | $PGD^{40}$ | $PGD^{100}$ | $CW^{\infty}$ | |
| TRADES | 99.48 | 96.07 | 95.52 | 95.69 | 98.12 | 97.86 | 98.21 | 92.79 |
| MART | 99.38 | 96.99 | 96.11 | 95.98 | 98.16 | 97.96 | 98.28 | 93.30 |
| AT | 99.41 | 96.01 | 95.49 | 95.78 | 98.05 | 97.73 | 98.20 | 88.50 |
| ATDA | 98.72 | 96.82 | 96.26 | 96.31 | 97.74 | 97.28 | 97.76 | 93.31 |
| $DIAL_{KL}$ (Ours) | 99.46 | 97.05 | 96.06 | 96.17 | 98.14 | 97.83 | 98.14 | **93.68** |
| $DIAL_{CE}$ (Ours) | **99.52** | **97.61** | **96.91** | **97.00** | **98.41** | **98.12** | **98.48** | 93.43 |

# E  ADDITIONAL RESULTS ON MNIST AND CIFAR-10

In Table 11 we present additional results using the PGD$^{1000}$ threat model. We use step size of 0.003 and constrain the attacks by the same perturbation $\epsilon = 0.031$. Table 12 presents a comparison of our method combined with AWP to other the variants of AWP that were presented in Wu et al. (2020). In addition, in Table 13 we add the $F_1$-robust scores for different variants of AWP.

Table 11: PGD$^{1000}$ attack on MNIST and CIFAR-10 on white-box and black-box settings.

| | MNIST | | CIFAR-10 | |
|---|---|---|---|---|
| Defense Model | White-box | Black-box | White-box | Black-box |
| TRADES | 95.22 | 97.81 | 56.43 | 83.80 |
| MART | 95.74 | 97.89 | 56.55 | 82.47 |
| AT | 95.36 | 97.78 | 54.40 | 83.96 |
| ATDA | 96.20 | 97.34 | 41.02 | 75.11 |
| DIAL$_{CE}$ (Ours) | **96.78** | **98.10** | 51.57 | **88.22** |
| DIAL$_{KL}$ (Ours) | 95.99 | 97.89 | **56.73** | 84.00 |

Table 12: Robustness comparison of DIAL-AWP and other variants of AWP that do not require additional data under the $\ell_\infty$ threat model.

| Defense Model | Natural | PGD$^{20}$ | PGD$^{100}$ | CW$_\infty$ | AA |
|---|---|---|---|---|---|
| DIAL-AWP (Ours) | **85.91** | **61.10** | **59.86** | **57.67** | **56.78** |
| TRADES-AWP (Wu et al., 2020) | 85.36 | 59.27 | 59.12 | 57.07 | 56.17 |
| MART-AWP (Wu et al., 2020) | 84.43 | 60.68 | 59.32 | 56.37 | 54.23 |
| AT-AWP (Wu et al., 2020) | 85.57 | 58.14 | 57.94 | 55.96 | 54.04 |

Table 13: $F_1$-robust measurement on AWP variants based on white-box attack.

| Defense Model | $F_1$-robust |
|---|---|
| DIAL-AWP (Ours) | **0.69753** |
| TRADES-AWP (Wu et al., 2020) | 0.68162 |
| MART-AWP (Wu et al., 2020) | 0.68857 |
| AT-AWP (Wu et al., 2020) | 0.67381 |

## F  EXTENDED RESULTS ON UNFORESEEN CORRUPTIONS

We present full accuracy results against unforeseen corruptions in Tables 14 and 15. We also visualize it in Figure 6.

Table 14: Accuracy (%) against unforeseen corruptions.

| Defense Model | brightness | defocus blur | fog | glass blur | jpeg compression | motion blur | saturate | snow | speckle noise |
|---|---|---|---|---|---|---|---|---|---|
| TRADES | 82.63 | 80.04 | 60.19 | 78.00 | 82.81 | 76.49 | 81.53 | 80.68 | 80.14 |
| MART | 80.76 | 78.62 | 56.78 | 76.60 | 81.26 | 74.58 | 80.74 | 78.22 | 79.42 |
| AT | 83.30 | 80.42 | 60.22 | 77.90 | 82.73 | 76.64 | 82.31 | 80.37 | 80.74 |
| ATDA | 72.67 | 69.36 | 45.52 | 64.88 | 73.22 | 63.47 | 72.07 | 68.76 | 72.27 |
| DIAL (Ours) | **87.14** | **84.84** | **66.08** | **81.82** | **87.07** | **81.20** | **86.45** | **84.18** | **84.94** |

Table 15: Accuracy (%) against unforeseen corruptions.

| Defense Model | contrast | elastic transform | frost | gaussian noise | impulse noise | pixelate | shot noise | spatter | zoom blur |
|---|---|---|---|---|---|---|---|---|---|
| TRADES | 43.11 | 79.11 | 76.45 | 79.21 | 73.72 | 82.73 | 80.42 | 80.72 | 78.97 |
| MART | 41.22 | 77.77 | 73.07 | 78.30 | 74.97 | 81.31 | 79.53 | 79.28 | 77.8 |
| AT | 43.30 | 79.58 | 77.53 | 79.47 | 73.76 | 82.78 | 80.86 | 80.49 | 79.58 |
| ATDA | 36.06 | 67.06 | 62.56 | 70.33 | 64.63 | 73.46 | 72.28 | 70.50 | 67.31 |
| DIAL (Ours) | **48.84** | **84.13** | **81.76** | **83.76** | **78.26** | **87.24** | **85.13** | **84.84** | **83.93** |

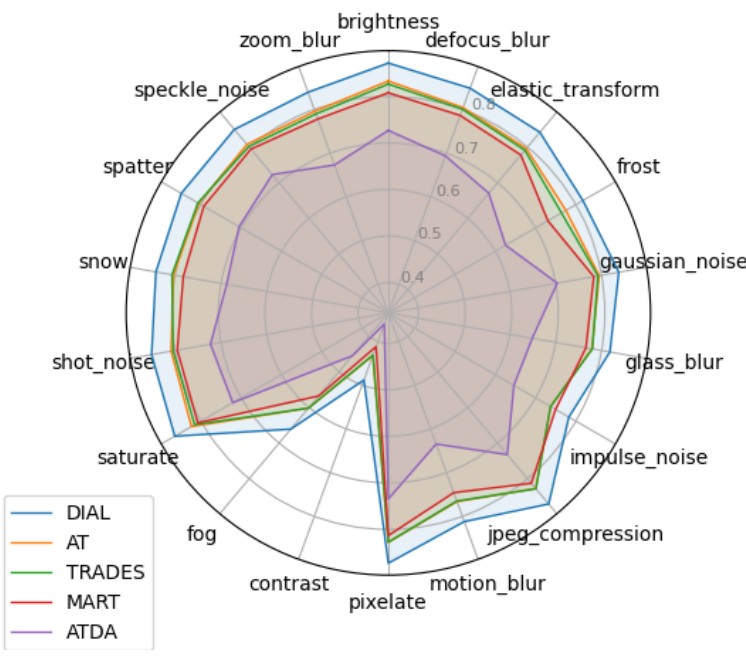

Figure 6: Accuracy comparison with all tested methods over unforeseen corruptions.

## G  TRANSFER LEARNING SETTINGS

The models used are the same models from previous experiments. We follow the common procedure of "fixed-feature" setting, where only a linear layer on top of the pre-trained network is trained. We train a linear classifier on CIFAR-100 on top of the pre-trained network which was trained on CIFAR-10. We also train a linear classifier on CIFAR-10 on top of the pre-trained nwork which was trained on CIFAR-100. We train the linear classifier for 100 epochs, and an initial learning rate of 0.1 which is decayed by a factor of 10 at epochs 50 and 75. We used SGD optimizer with momentum 0.9.

## H    EXTENDED VISUALIZATIONS

In Figure 8, we provide additional visualizations of the different adversarial training methods presented above. We visualize the models outputs using t-SNE with two components on the natural test data and the corresponding adversarial test data generated using PGD[20] white-box attack with step size 0.003 and $\epsilon = 0.031$ on CIFAR-10.

In Figure 7 we visualize statistical differences between natural and adversarial examples in the feature representation layer. Specifically, we show the differences in mean and std on thirty random feature values from the feature representation layer as we pass through a network the natural test examples and their corresponding adversarial examples. We present the results on same network architecture (WRN-34-10), trained using three different training procedures: naturally trained network, network trained using standard adversarial training (AT) Madry et al. (2017), and DIAL on the CIFAR-10 dataset. When the statistical characteristics of each feature differ from each other, it implies that the features layer is less domain invariant. That is, smaller differences in mean/std yields a better invariance to adversarial examples. One can observe that for DIAL, there is almost no differences between the mean/std of natural examples and their corresponding adversarial examples. Moreover, for the vast majority of the features, DIAL present smaller differences compared to the naturally trained model and the model trained with standard adversarial training. Best viewed in colors.

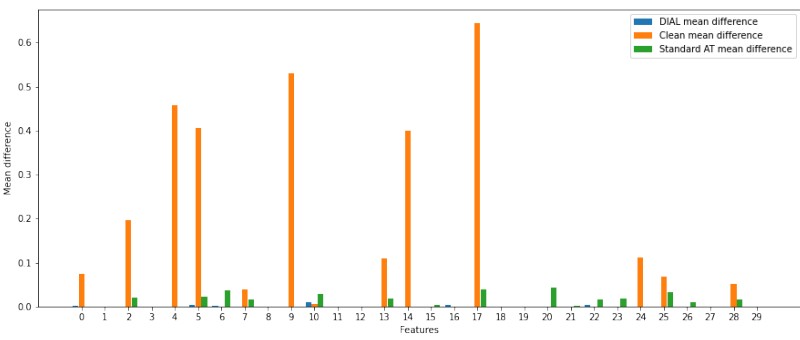

(a) Mean difference comparison

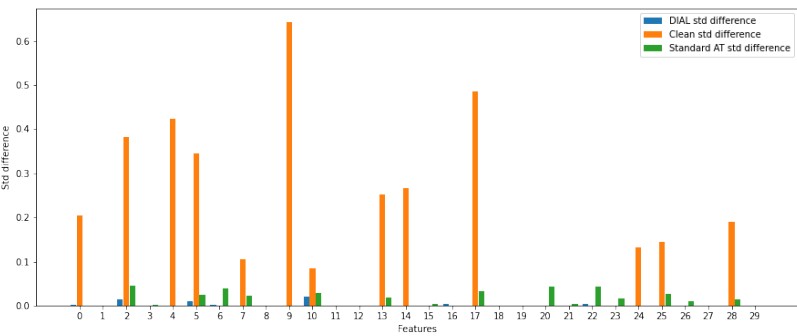

(b) Standard deviation difference comparison

Figure 7: Mean and std differences comparison between DIAL, naturally trained model and model trained using standard adversarial training on thirty random features from the features layer on the CIFAR-10 dataset with WRN-34-10 architecture. Each bar represents the absolute difference between the means/std of the natural examples and the mean/std of their corresponding adversarial examples on this same feature.

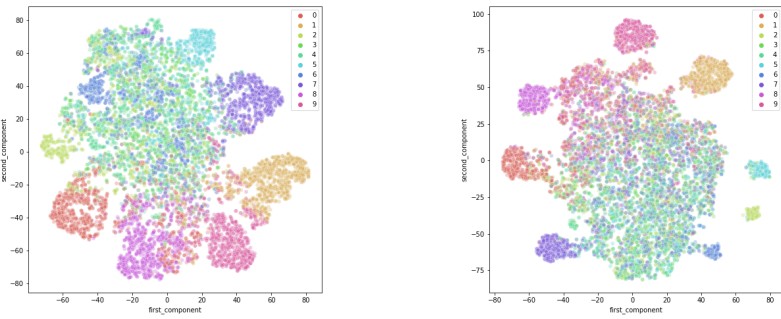

(a) **MART** embedded model output on natural test data

(b) **MART** embedded model output on adversarial test data

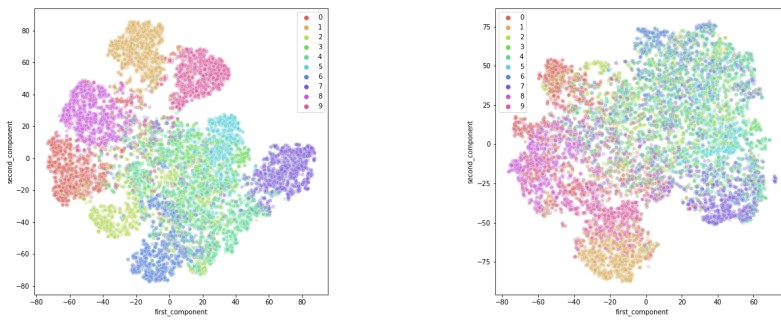

(c) **AT** embedded model output on natural test data

(d) **AT** embedded model output on adversarial test data

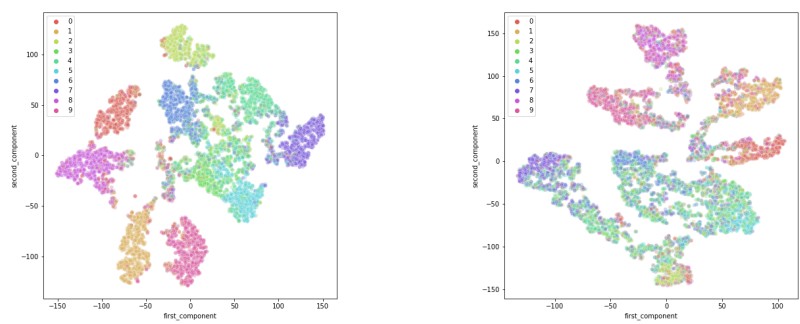

(e) **ATDA** embedded model output on natural test data

(f) **ATDA** embedded model output on adversarial test data

Figure 8: t-SNE embedding of model output in two-dimensional space for MART, AT, and ATDA under natural and adversarial test data from CIFAR-10.

