# OpenReview forum: "Domain Invariant Adversarial Learning"
_ICLR.cc/2022/Conference — ICLR 2022 Submitted_

### Official Review · Reviewer_FjGF · 2021-10-30

**Correctness:** 4
**Technical Novelty And Significance:** 3
**Empirical Novelty And Significance:** 4
**Recommendation:** 5
**Confidence:** 4

**Main Review:**

This paper proposes a simple and effective adversarial learning method DIAL, which brings the idea from domain adaptation for robust representation.

Strengths:
1. This paper is well-written and easy to follow.
2. It conducts various experiments to demonstrate the effectiveness of the proposed method ranging from robustness to white-box attacks, black-box attacks, unforeseen adversaries, unforeseen corruptions and transfer learning. The experimental results are solid and technically sound.

Weaknesses:
1. From my point of view, the novelty of the methodology is not enough, as the domain classifier and the gradient reversal layer are the same with those methods in domain adaptation such as [1].
2. To better understanding the reversal-ratio hyper-parameter $r$, can the authors provides the robustness under different values of $r$.

[1] Yaroslav Ganin and Victor Lempitsky. Unsupervised domain adaptation by backpropagation. In International conference on machine learning, pp. 1180–1189. PMLR, 2015.




**Summary Of The Paper:**

This paper proposes a domain invariant adversarial training (DIAL) method, which learns the feature representation that is both robust and domain invariant.  Apart from the label classifier, the model is equipped with a domain classifier that constrains the model not to discriminate between natural examples and adversarial examples, thus achieving a more robust feature representation. Extensive experiments on image classification benchmark the robustness compared to other state-of-the-art methods.

**Summary Of The Review:**

Overall, this paper proposes a simple and effective adversarial learning method DIAL for robust representation learning. Extensive experiments are conducted to demonstrate the effectiveness of the proposed method and provide solid results. However, the novelty of the paper is not significant as similar methodology exists in domain adaptation.

---

> ### Author Response · Authors · 2021-11-10
> **Response**
>
> Thank you very much for your comments.
> We indicate the comments/questions by Q and our answers by A.
>
> Q1: From my point of view, the novelty of the methodology is not enough, as the domain classifier and the gradient reversal layer are the same with those methods in domain adaptation such as [1].
>
> A1: Let us stress that while the *techniques* we use are standards, these are used in a novel setting and, as the experiments indicate, outperform the state of the art. We gave an explanation about the novelty and simplicity of DIAL in a general comment. We hope this explanation will help in addressing your concerns.
>
> Q2: To better understand the reversal-ratio hyper-parameter r, can the authors provide the robustness under different values of r.
>
> A2: Thank you for this important suggestion.
> We attach results on several reversal-ratio values for DIAL_KL and DIAL_CE.
> As can be seen by the tables below, each method has its own reversal ratio. This is aligned with the magnitude and different losses in KL and CE. Moreover, the reversal ratio increases both robust and natural accuracy compared to no reversal-ratio.
>
> For DIAL_CE - as we increase r up to 0.1 both robust and natural accuracy increases, after 0.1, natural accuracy slightly decreases while the robust accuracy keeps increasing.
> For DIAL_KL - as we increase r up to 4 both robust and natural accuracy increases. from 2, natural accuracy  slightly decreases while the robust accuracy keeps increasing.
>
> These observations are also in line with the fact that as we increase the influence of our DA head, the representation will be more domain invariant, which in turn will lead to better robustness.
>
> DIAL-CE              | r=0    | r=0.1  | r=1     | r=2
>
> Robust accuracy | 51.82 | 54.31 | 54.35 | 54.36 |
>
> Natural accuracy | 87.77 | 89.59 | 88.51 | 88.34 |
>
>
>
>
> DIAL-KL              | r=0     | r=1     |r=2      |r=3     | r=4    |
>
> Robust accuracy | 56.60 | 57.61 | 57.90 | 58.20 | 58.43 |
>
> Natural accuracy | 84.92 | 86.28 | 85.96 | 85.63 |85.25 |

---

> ### Author Response · Authors · 2021-11-28
> **Kind reminder about feedback**
>
> Dear reviewer FjGF ,
>
> We are very grateful for your initial review and  insightful suggestion about the reversal-ratio hyper-parameter experiment we have added. As the discussion phase end tomorrow, we will appreciate if you could check our response, as well as the new experiment results and our general comment about novelty, and let us know if it addressed your concerns. We would be eager to answer any other concern you might have.
>
> Thank you very much!
>
> -- Authors

---

### Official Review · Reviewer_FZTb · 2021-10-31

**Correctness:** 3
**Technical Novelty And Significance:** 2
**Empirical Novelty And Significance:** 3
**Recommendation:** 3
**Confidence:** 4

**Main Review:**

Pros:

(1) The paper is clearly written and easy to follow.

(2) The motivation behind is very intuitive.

(3) The paper conducts extensive experiments including multiple-$\ell_p$-norm adversarial perturbations and unseen corruptions.

Cons:

(1) My biggest concern is the novelty of this paper. Though showing promising performance, the idea of learning a feature extractor to minimize the distance between adversarial and clean distributions/domains has been widely studied and adopted before in the domain adaptation (DA) literature. In this paper, the author just simply introduced several DA loss terms and used the GAN framework to learn a more robust model. The experimental results are persuasive, however, the approach is too simple and not novel enough.

(2) Some minor problems. I cannot find your paper in the autoattack leaderboard as you mentioned at the first line in Page 7.

**Summary Of The Paper:**

This paper proposes DIAL to learn domain-invariant representations for clean and adversarial examples to improve model robustness and clean accuracy. The main idea is to treat the problem as a domain adaptation problem by considering the data shift between adversarial and clean distributions, and then use the generative adversarial network (GAN) principle to tackle this data shift.

**Summary Of The Review:**

I think the paper conducts extensive experiments to demonstrate the effectiveness of the proposed method, including some interesting ones, e.g., robustness against unseen perturbations, transfer learning (I like them). However, the novelty of this paper is insufficient, and using the domain adaptation principle and learning invariant representation has been widely studied. Therefore, I vote for rejection.

---

> ### Author Response · Authors · 2021-11-10
> **Response**
>
> Thank you for acknowledging the persuasive nature of our experiments, and for your valuable comments.
> We indicate the comments/questions by Q and our answers by A.
>
>
> Q1: My biggest concern is the novelty of this paper. Though showing promising performance, the idea of learning a feature extractor to minimize the distance between adversarial and clean distributions/domains has been widely studied and adopted before in the domain adaptation (DA) literature. In this paper, the author just simply introduced several DA loss terms and used the GAN framework to learn a more robust model. The experimental results are persuasive, however, the approach is too simple and not novel enough.
>
> A1: In a general comment, we have argued that simplicity is a virtue rather than a drawback, and also addressed the novelty concerns.
> Let us stress that while the techniques we use are standards, these are used in a novel setting and, as the experiments indicate, outperform the state of the art. We hope this explanation will help in addressing your concerns.
>
> Q2: Some minor problems. I cannot find your paper in the auto-attack leaderboard as you mentioned at the first line in Page 7.
>
> A2: You are right, we did not submit the model to the benchmark yet since we did not want to break the double-blind review. We will submit the model after the review process is done. The results are of course reproducible using the supplied code.

---

> ### Author Response · Authors · 2021-11-28
> **Kind reminder about feedback**
>
> Dear reviewer FZTb,
>
> We wish to thank you again for your initial review. As the discussion phase end tomorrow, we would be grateful if you could check our response, as well as the general comment on novelty, and let us know if it addressed your concerns. We would be eager to answer  any other concern you might have.
>
> Thank you!
>
> -- Authors

---

### Official Review · Reviewer_8wyr · 2021-11-02

**Correctness:** 4
**Technical Novelty And Significance:** 2
**Empirical Novelty And Significance:** 2
**Recommendation:** 3
**Confidence:** 3

**Main Review:**

+ The paper is easy to follow and the idea is straightforward.
- The experiment section is not comprehensive. Only a few methods are included in the comparison. More recent SOTA methods are missing.

**Summary Of The Paper:**

In this paper, DANN is leveraged to generate domain invariant and robust feature representation. The authors claim that the proposed method outperforms other methods when the target domain is the adversarial examples.


**Summary Of The Review:**

+ The paper is easy to follow and the idea is straightforward.
- The experiment section is not comprehensive. Only a few methods are included in the comparison. More recent SOTA methods are missing.

---

> ### Author Response · Authors · 2021-11-10
> **Response**
>
> Thank you for your comments and suggestions.
> We indicate the comments/questions by Q and our answers by A.
>
>
> Q1: The experiment section is not comprehensive. Only a few methods are included in the comparison. More recent SOTA methods are missing.
>
> A1: We thank the referee for his/her comments.
> We took great pains to review the SOTA methods in the literature, and chose the ones that exhibited the strongest performance (TRADES, MART, AWP, etc.) in Tables 3-7. We compare ourselves against the methods from the recent papers appearing in top-tier conferences. We did not, however, include the techniques that were later broken by strong adversaries (e.g., feature-scatter, adversarial interpolation training, etc.) as was shown in [1].
>
> [1] Reliable evaluation of adversarial robustness with an ensemble of diverse parameter-free attacks
>
> We wish this comment were more detailed, so that we could address the specific points. Which additional experiments would the referee like to see?

---

> ### Author Response · Authors · 2021-11-28
> **Kind reminder about feedback**
>
> Dear reviewer 8wyr,
>
> We wish to thank you again for your initial review. As the discussion phase end tomorrow, we would be grateful if you could check our response and let us know if it addressed your concerns. We would be eager to answer any other concerns you might have.
>
> Thank you,
>
> -- Authors

---

### Official Review · Reviewer_iRX5 · 2021-11-03

**Correctness:** 4
**Technical Novelty And Significance:** 2
**Empirical Novelty And Significance:** 3
**Recommendation:** 8
**Confidence:** 4

**Main Review:**

**Strengths:**

* The idea is simple, yet it leads to significantly more robust networks
* The paper is well written, and it is easy to follow
* While the experiments are only carried out on smaller scale datasets, they are thorough, and they support the claims of the authors


**Weaknesses:**

I don't see major weaknesses in the paper. Below are some minor points.

* DIAL-AWP comes out of the blue in Table 3. For the sake of consistency, I suggest adding it to Tables 1 and 2 as well and providing the formulation (for self-sufficiency).

* The TSNE plots in Figure 5 for clean and perturbed distributions seem to have been calculated separately, which means that we are effectively looking at two different embedding spaces when we look at (a) and (b). I suggest that the author append the clean and perturbed representations, calculate the TSNE embedding jointly, and then plot them into their corresponding plots.

**Additional Comments/Questions:**

* In your KL robustness loss you have,
 $$ \mathcal{L}_{rob}^{KL}=\frac{1}{n}\sum_i KL(G_f(x'_i;\theta_f)||G_f(x_i;\theta_f))$$
My understanding is that $G_f$
 is your feature extractor, and $G_f(x'_i), G_f(x_i)\in \mathbb{R}^d$  are not probability vectors, this is while $KL(\cdot||\cdot)$ is a dissimilarity measure defined only for probability distributions. Could you comment on this? Also, wouldn't a simple MSE work fine here?

* This might be a matter of style, but it could be helpful to add equation numbers to your equations.

* Typos:

  * Page 3 second paragraph: "belongs to the the family"
  * Page 5 second to the last paragraph: "the initial learinnig"


**Summary Of The Paper:**

The paper describes an adversarial training approach that, in addition to the commonly used robustness loss, requires the network to extract similar representation distributions for clean and attacked data. The proposed method is inspired by domain adaptation approaches that require a model to extract domain invariant/agnostic features from two domains. In the context of this paper, the two domains are the clean and adversarially perturbed images, and the network is required to extract domain invariant representation. To achieve domain invariance, the authors propose a domain classifier ( i.e., an adversarial network) that discriminates the representations from clean and attacked images. The feature extractor is then required to generate features that fool the domain classifier. The authors then provide extensive experiments on small-scale benchmark datasets (SVHN, CIFAR10, CIFAR 100, and MNIST in the supplementary material ) to show the robustness of their proposed approach against the state-of-the-art robustness methods under white-box and black-box attacks. The authors show that their proposed method provides: 1) higher accuracy on attacked data (more robustness), and 2) higher accuracy on clean data, closing the gap between the performance on clean and attacked data. In addition, the paper provides insightful experiments on robustness to unforeseen adversaries, robustness to unforeseen corruptions, transfer learning, and ablation studies.



**Summary Of The Review:**

**Overall assessment:** The paper is well-written and easy to follow. While the main idea of learning domain invariant features is simple, its use in the context of robustness against adversarial attacks seems to lead to a significant performance boost. The experiments and, in particular, the ablation study section is insightful and aligned with the paper's claims. I think the paper is above average, and therefore I would like to vote for its acceptance.

---

> ### Author Response · Authors · 2021-11-10
> **Response**
>
> We are grateful for this detailed, comprehensive review. We will respond to the points raised in detail.
> We indicate the comments/questions by Q and our answers by A.
>
> Q1: DIAL-AWP comes out of the blue in Table 3. For the sake of consistency, I suggest adding it to Tables 1 and 2 as well and providing the formulation (for self-sufficiency)
>
> A1: Thank you for suggesting this, we are working on adding DIAL-AWP to Tables 1 and 2 as you suggested in the revised version.
>
> Q2: The TSNE plots in Figure 5 for clean and perturbed distributions seem to have been calculated separately, which means that we are effectively looking at two different embedding spaces when we look at (a) and (b). I suggest that the author append the clean and perturbed representations, calculate the TSNE embedding jointly, and then plot them into their corresponding plots
>
> A2: Thank you for this suggestion, we calculated the TSNE embedding jointly as you suggested, we will also add it to the revised version. We uploaded the updated plots in the following links: https://ibb.co/Np9CfKK (tsne-adv-dial), https://ibb.co/sHncPvP (tsne-adv-trades), https://ibb.co/HxsJNGs (tsne-clean-dial), https://ibb.co/qB1fFph (tsne-clean-trades)
>
> Q3: My understanding is that  G_f is your feature extractor G_f(x), G_f(x’), are not probability vectors, this is while KL(.||.) is a dissimilarity measure defined only for probability distributions. Could you comment on this? Also, wouldn't a simple MSE work fine here?
>
> A3: You are right, this is a mistake in the formula . The correct formula is KL(G_y(G_f(x)) || G_y(G_f(x))), where a softmax is applied on G_y(G_f(.)) to yield a probability vector.
> About MSE, we actually tried to apply MSE on the features (G_f(.)) representation, but it did not show better results compared to the KL.
>
> Q4: Typos
>
> A4: Thank you for pointing out these typos, we will fix them in the revision.

---

### Author Response · Authors · 2021-11-10
**A general comment about the novelty/simplicity**

We should have been clearer about our conceptual contribution. In essence, we exhibit a reduction from robust learning to domain adaptation.

While the Ganin et al. (DANN) approach would indeed be standard in a domain adaptation setting, we argue that our novel application in robustness required overcoming several non-trivial challenges. Our insight consists of constructing a specific pipeline that combines adversarial training with DANN.

On the one hand, in domain adaptation, DANN is applied to distinct domains (e.g., a cat dataset vs. a dog dataset). On the other hand, in adversarial training, there are no specific distinct domains. Rather, the adversarial training process dynamically generates adversarial examples. Then, we simultaneously combine DANN objectives with adversarial training objectives. This ensures that adversarial examples will lie on the same manifold as the natural examples, leading to a more robust representation.

In contradistinction with domain adaptation, where the objective is to find a good representation across different distributions, in our setting we seek to optimize the conflicting objectives of accuracy (on natural examples) and robustness (on adversarial examples) -- on the same distribution. We reduce the problem of robust learning to one of domain adaptation by implicitly constructing the two distributions.

Additionally, we argue that the simplicity of our approach is a benefit rather than a drawback: when a simple approach proves effective, this greatly facilitates its implementation and broad adaptation.

---

> ### Author Response · Authors · 2021-11-26
> **A gentle reminder**
>
> Dear reviewers,
>
> As the final discussion phase is comming to an end, we would like to know if there are any further concerns/questions/suggestions we can adress to?
>
> We would be pleased to answer and discuss with you about any additional concern you might have.
>
> Best regards,
>
> -- Authors

---

### Decision · Program_Chairs · 2022-01-20

**Decision:**

Reject

**Comment:**

The paper describes an adversarial training approach that, in addition to the commonly used robustness loss, requires the network to extract similar representation distributions for clean and attacked data. The proposed method is inspired by domain adaptation approaches that require a model to extract domain invariant/agnostic features from two domains. Although the experimental results are solid and technically sound, the novelty of the methodology is not enough, as the domain classifier and the gradient reversal layer are the same with those methods in domain adaptation such as "unsupervised domain adaptation by backpropagation". On the other hand, more recent SOTA methods are missing and only smaller scale datasets are used for evaluation. During the discussions, the major concerns from three reviewers are novelty.

I totally agree that the simplicity of the method should be a virtue. However, the idea of domain-invariant representation learning is already established well, and its application to adversarial training is quite intuitive to the community. Also, the similar methodology already exists in domain adaptation. According to the top-tier conference culture in the ML community, what most valuable is the novelty and insight, not the performance. In the end, I think that this paper may not be ready for publication at ICLR, but the next version must be a strong paper.